# Energy Drink Consumption and Substance Use among Middle and High School Students

**DOI:** 10.3390/ijerph17093110

**Published:** 2020-04-29

**Authors:** Hugues Sampasa-Kanyinga, Lydie Masengo, Hayley A. Hamilton, Jean-Philippe Chaput

**Affiliations:** 1School of Epidemiology and Public Health, University of Ottawa, Ottawa, ON K1G 5Z3, Canada; jpchaput@cheo.on.ca; 2Healthy Active Living and Obesity Research Group, Children’s Hospital of Eastern Ontario Research Institute, Ottawa, ON K1H 8L1, Canada; 3Department of Psychology, Carleton University, Ottawa, ON K1S 5B6, Canada; lydiemasengo@cmail.carleton.ca; 4Institute for Mental Health Policy Research, Centre for Addiction and Mental Health, Toronto, ON M5S 2S1, Canada; hayley_hamilton@camh.net; 5Dalla Lana School of Public Health, University of Toronto, Toronto, ON M5T 3M7, Canada

**Keywords:** energy drinks, drug use, sex, middle school, high school, adolescents

## Abstract

This study examined the association between energy drink consumption and substance use among adolescents and tested whether sex and/or grade level (i.e., middle vs. high school) moderate the association. Data were derived from the 2017 Ontario Student Drug Use and Health Survey, a representative survey of students in 7th to 12th grade. Analyses included 10,662 students who self-reported information on energy drink consumption and substance use. Poisson regression models were used with adjustments for important covariates. Energy drink consumption was associated with tobacco cigarette smoking (incidence rate ratio (IRR): 3.74; 95% confidence interval (CI): 3.22–4.35), cannabis use (IRR: 2.90; 95% CI: 2.53–3.32), binge drinking (IRR: 2.46; 95% CI: 2.05–2.96), opioid use (IRR: 2.23; 95% CI: 1.85–2.68), and alcohol use (IRR: 1.31; 95% CI: 1.26–1.36). The associations of energy drink consumption with tobacco cigarette smoking, cannabis use, and alcohol consumption were modified by grade level (two-way interaction terms *p* < 0.05). The association between energy drink consumption and substance use was generally much stronger among middle school students compared with high school students. The findings suggest that middle school students may be more vulnerable to the negative effects of energy drinks in relation with substance use.

## 1. Introduction

Energy drinks are highly caffeinated beverages that have targeted marketing to adolescents, with the claim that these products give an extra boost in energy, alertness, and mental performance [1,2,3,4]. However, their use has been associated with negative health effects in this age group, such as dehydration, heart complications (arrhythmia, health failure), anxiety, and insomnia [5]. The effects of energy drinks are theoretically attributed to the high caffeine content, which can range anywhere from 50 mg to as high as 505 mg per can or bottle [6]. Small amounts of caffeine can have a greater effect on children because of their smaller body size and the fact that their brain is still under development [7]. Caffeine impacts brain development by its antagonist effect on adenosine receptors contained in the brain [7]. Although Health Canada and the American Academy of Pediatrics recommend that children and adolescents should never consume energy drinks [8,9], research has shown that consumption of these beverages are common among adolescents. For example, a recent Canadian report indicated that nearly one-third (34.1%) of students in grades 7 through 12 report drinking an energy drink at least once in the past year [10]. The ease of availability of energy drinks, increasing popularity among adolescents, and numerous associated health-compromising behaviors among adolescent consumers have rendered energy drink consumption an important public health issue [3,6,11]. This makes the relationship between energy drink consumption and health indicators an important area of study.

Evidence from past research studies has shown that energy drink consumption among adolescents is associated with substance use, such as alcohol, tobacco cigarette, cannabis, and nonmedical use of prescription drugs [12,13,14], suggesting that substance use behaviors tend to cluster within individuals. For example, Azagba et al. [15] found that involvement in risky behaviors including tobacco, alcohol, marijuana, and other drugs and substances was higher among consumers of energy drinks relative to non-users, in a sample of Canadian students. Likewise, in a sample of USA adolescents, Terry-McElrath et al. [16] found that the frequency of energy drink consumption was significantly and positively linked with the frequency of substance use in the past 30 days, such as alcohol, cigarettes, marijuana, and amphetamines. Energy drinks are commonly mixed with alcohol because of their high caffeine content, often in an attempt to reduce feelings of alcohol intoxication [17,18]. This may increase the potential for alcohol-related injury. Seifert et al. [19] found that males reported usually consuming energy drinks with alcohol or drugs, whereas females reported consuming energy drinks with other pharmaceuticals.

Research studies have indicated important variations in energy drink consumption or substance use by grade and sex. High school students have been found to consume more energy drinks [20,21,22,23] and report substance use than their middle school counterparts. Male adolescents are more likely than females to consume energy drinks [15,23,24,25]. Sex differences were also found in the number of emergency room visits involving energy drinks [5,26]. More males visited than females, which puts young males at a greater risk of adverse consequences of energy drink consumption. Boys are also well known to be more likely than girls to engage in substance use. Given the above-mentioned sex and grade differences in both energy drink consumption and substance use among adolescents, it is possible that the relationship between energy drink consumption and substance use differs between middle and high school students, as well as between males and females. However, there still remains a lack of research investigating whether sex and grade level could moderate the link between energy drink consumption and substance use among adolescents. Previous research has either focused on one grade level (i.e., middle school only [27] or high school students only [15]) or combined both middle and high school students [13,14,16] in their analyses. Using middle school students in relation to high school students is important for several reasons. First, it could help examine whether the associations between energy drink consumption and substance use vary between middle and high school students. Second, it can help to inform future intervention programs on whether or not they should start early on by targeting middle school students rather than their high school counterparts. Indeed, current school intervention programs related to energy drink consumption mostly focus on high school students due to the high prevalence of consumption in these grade levels [28]. However, energy drink consumption in younger students may be more devastating because their brain is still under development [7]. Finally, it will inform future research regarding the need or not to be conducted using middle school students in relation to high school students, rather than having them combined, and by generating hypotheses in relation to other health risk behaviors.

Thus, the present research aims to shed light into the moderating effects of sex and grade level on the relationship between energy drink consumption and substance use among adolescents. Investigating this issue can help to inform the development of appropriate and tailored public health interventions.

## 2. Materials and Methods

### 2.1. Participants

Data were derived from the 2017 cycle of the Ontario Student Drug Use and Health Survey (OSDUHS), a representative province-wide survey of 11,435 students in grades 7 through 12 in publicly funded schools, representing about 93% of the province’s adolescent population [10]. Excluded from selection were schools on military bases, in First Nations communities, hospitals and other institutions, and private schools. Special education classes and English as a second language classes were also excluded from selection. The OSDUHS has been conducted every two years since 1977 with the current survey focused on investigating student drug use, mental health, physical health, gambling, bullying, and other risky behaviors, as well as identifying risk and protective factors. This self-administered, anonymous survey uses a stratified (region by school level) two-stage (school, class) cluster design and involved 11,435 students from 764 classes in 214 schools in the 2017 cycle. The participation rate among students was 61%, with non-participants consisting of students who were absent (12%) and those with parental refusal or unreturned consent forms (27%). Participation was considered above average for a student survey where active consent from a parent or guardian is necessary [29,30]. The 2017 OSDUHS was approved by the Research Ethics Boards at the Centre for Addiction and Mental Health and York University, as well as 31 school board research review committees. Student participation required parental consent for those aged under 18 years, as well as student assent. The survey was self-administered, anonymous, and took approximately 30 min to complete. To maximize validity and to enhance cross-study comparability, many of the OSDUHS questionnaire items were derived from international guidelines and recognized student surveys, such as NIDA’s Monitoring the Future (MTF) survey, the CDC’s Youth Risk Behavior Survey (YRBS), and the WHO’s Health Behavior in School-aged Children (HBSC) survey. Furthermore, the survey uses validated scales and screeners. All newly introduced items in the 2017 questionnaire were evaluated by both expert review and pretested on a small convenience sample of young adolescents. The readability of the 2017 questionnaire showed a 7th-grade reading level according to the Flesch-Kincaid reading score and the evaluation of the comprehension and sensitive nature of the questionnaire showed a positive assessments, with 97% of students (96% of 7th graders) indicating that the questionnaire was “fairly” or “very easy” to understand. Detailed information on the survey design and methods are described elsewhere [10].

### 2.2. Measures

#### 2.2.1. Outcome Variables

Alcohol, tobacco, and cannabis use were measured with the following three questions: “In the last 12 months, how often did you drink alcohol (liquor, wine, beer, coolers)?”; “In the last 12 months, how often did you smoke cigarettes?” and “In the last 12 months, how often did you use cannabis (e.g., “marijuana”)?” For purposes of analyses, responses were binary coded indicating use at least once vs. non-use during the last 12 months. A few puffs for tobacco cigarette and a sip for alcohol were included among non-use during the last 12 months. A sensitivity analysis contrasting a more severe level of substance use (at least 2 or 3 times a month of alcohol consumption, at least 3 to 5 cigarettes a day, and 6 to 9 times of cannabis use in the last 12 months) to non-regular substance use as the outcome was run. Binge drinking was measured through a question asking how often students had 5 or more drinks of alcohol on the same occasion (i.e., binge drinking) during the past 4 weeks. Responses were grouped to create a binary measure that reflected any binge drinking versus none.

Use of nonmedically prescribed opioids in the last 12 months was measured with the following question: “In the last 12 months, how often did you use pain relief pills (such as Percocet, Percodan, Tylenol #3, Demerol, OxyNeo, Oxycontin, Codeine) without a prescription or without a doctor telling you to take them? (We do not mean regular Tylenol, Advil, or Aspirin that anyone can buy in a drugstore)”. Response options were 1 or 2 times; 3 to 5 times; 6 to 9 times; 10 to 19 times; 20 to 39 times; 40 or more times; used without a prescription but not in the last 12 months; never used without a prescription in lifetime; and don’t know what pain relief pills are. The three latter response options were combined to reflect “no use of nonmedically prescribed opioids in the last 12 months”, contrasting with “use of nonmedically prescribed opioids at least once in the last 12 months.” In a sensitivity analysis, the use of nonmedically prescribed opioids for at least 3 to 5 times in the last 12 months was contrasted to non-regular opioid use (i.e., no use or 1 to 2 times) in the last 12 months.

#### 2.2.2. Independent Variable

Energy drink consumption was measured by the following item: “In the last 7 days, how often did you drink a can of a high-energy caffeine drink, such as Red Bull, Rockstar, Amp, Full Throttle, Monster, etc.?”. Response categories referred to consumption (differentiating between “1 time”, “2 to 4 times”, “5 to 6 times”, “once each day”, or “more than once each day”); no consumption in the last 7 days but some consumption in the last 12 months; and no consumption in the last 7 days or in the last 12 months. A dichotomous measure was constructed to represent “consumption” and “non-consumption” of energy drinks in the last 12 months.

#### 2.2.3. Potential Moderator

Sex (male/female) and grade level (middle school/high school) were used as potential moderators in our analyses. Grade level was constructed by collapsing grades 7 and 8 to represent the “middle school” level and grades 9 through 12 to represent the “high school” level [10].

#### 2.2.4. Covariates

Sociodemographic characteristics included age (years), sex (male/female), ethnicity (White, Black, East/South-East Asian, South Asian, Other), and subjective socioeconomic status (SES). SES was based on the youth version of the MacArthur Scale of Subjective Social Status [31], which was slightly modified to assess the family’s place within society. A ladder of 10 rungs was drawn and respondents were asked to place an “X” on the rung on which they feel their family stands relative to other families, based on SES indicators, including money, education, and jobs. Screen time was measured using an item that asked students to report the average number of hours they spent in a day watching TV/movies, playing video/computer games, chatting on a computer, emailing, or surfing the Internet in the last 7 days. Response options were “none,” “≤1 h/day,” “1–2 h/day,” “3–4 h/day,” “5–6 h/day,” and “≥7 h/day.” Responses of ≤2 h/day corresponded to students that met the screen time guideline recommendation. Remaining responses corresponded to students that did not meet the recommendation.

### 2.3. Statistical Analyses

All statistical analyses were performed using STATA (version 14.0, Stata Corp., College Station, Texas, USA). Analyses included participants with complete information on all variables, reducing the sample size from 11,435 to 10,662. Excluded participants did not differ from those included in our analyses for all the variables. Taylor series methods were used to compute unbiased variances, standard errors, and point estimates given the complex sample design of the OSDUHS. The estimation model was based on a design with 18 strata (region by school level) and 195 primary sampling units (schools). Weights were used within the analyses to adjust for the unequal probability of selection. Descriptive statistics of participants by sex and grade level were compared by a Pearson Chi-Square adjusted for the survey design and transformed into an F-statistic for categorical data and by an adjusted Wald test for continuous data. Following the recommendation to estimate relative risk for a common (i.e., prevalence of 10 % or more) binary outcome [32,33], Poisson regression with a robust variance estimator has been proposed to estimate the relative risk for a common dichotomous outcome [32]. Thus, we used univariate (Model 1) and multivariate (Model 2) Poisson regression analyses to examine the association between energy drink consumption and outcome variables of alcohol consumption, tobacco cigarette smoking, cannabis use, binge drinking, and opioid use. Covariates included age, sex, ethnicity, subjective SES, and screen time. Screen time could be a common cause of energy drink consumption and substance use, particularly with content exposure and marketed strategies to youth on new media [34,35,36], and was therefore added as a covariate. In order to test if the associations between energy drink consumption and substance use vary by sex and grade level, two-way interactions were examined in separate models for sex (Model 3) and grade level (Model 4). Grade level was a significant moderator of the association between energy drink consumption and substance use. Thus, subsequent analyses examining the association between energy drink consumption and substance use were stratified by grade level. There was no collinearity between age and grade.

## 3. Results

The descriptive characteristics of the sample are shown in Table 1. Over 70% of the students were in high school, almost one half were female, and 55% identified themselves as White. The average age of the sample was 15.1 years. One third of adolescents (33%) met the screen time recommendation. Middle school students were more likely than high school students to meet the screen time recommendation and to report higher subjective SES. High school students were significantly more likely to report alcohol use, tobacco cigarette smoking, and cannabis use within the past year, as well as binge drinking in the past month, compared to their middle school counterparts. Nearly one third of adolescents reported that they consumed at least one energy drink in the past 12 months. Males were more likely than females to report energy drink consumption in the past 12 months. High school students were also more likely than those in middle schools to report energy drink consumption in the past 12 months (38% vs. 24%, *p* < 0.001).

Bivariate associations between energy drink consumption and substance use are outlined in Table 2. Energy drink consumption in the past 12 months was more prevalent among individuals who did not meet the screen time recommendations compared to those who met the recommendation. It was also more prevalent among students who used alcohol, tobacco, cannabis, and nonmedical prescription opioids, and who reported binge drinking, than among adolescents who did not engage in such behaviors.

Regression analyses focused only on energy drink consumption in the past 12 months. Table 3 presents the results from Poisson regression analyses examining the associations between energy drink consumption in the past 12 months and substance use among adolescents.

Energy drink consumption was strongly associated with tobacco cigarette smoking (incidence rate ratio (IRR): 3.74; 95% confidence interval (CI): 3.22–4.35), cannabis use (IRR: 2.90; 95% CI: 2.53–3.32), binge drinking (IRR: 2.46; 95% CI: 2.05–2.96), opioid use (IRR: 2.23; 95% CI: 1.85–2.68), and alcohol use (IRR: 1.31; 95% CI: 1.26–1.36). The associations of energy drink consumption with tobacco cigarette smoking, cannabis use and alcohol consumption were modified by grade level (two-way interaction terms *p* < 0.05). Sex was not a significant moderator of the associations between energy drink consumption and substance use.

Results from multivariate Poisson regression analyses examining the associations between energy drink consumption in the past 12 months and substance use among adolescents stratified by grade level are presented in Table 4. After adjusting for covariates, middle school students who consumed energy drinks at least once in the past 12 months had stronger risks (compared to their high school counterparts) of smoking tobacco cigarette (IRR = 7.97 vs. IRR = 3.54), cannabis use (IRR = 10.27 vs. IRR = 2.73), and alcohol consumption (IRR = 1.43 vs. IRR = 1.28). There were no significant differences between middle and high school students in the association between energy drink consumption and opioid use. Energy drink consumption was not associated with binge drinking among middle school students before and after adjusting for covariates.

## 4. Discussion

### 4.1. Summary of Key Findings

This study examined the association between energy drink consumption and substance use in a large and representative sample of adolescents and tested whether sex and grade level would moderate these relationships. Energy drink consumption was strongly associated with alcohol use, tobacco cigarette smoking, cannabis use, binge drinking, and opioid use before and after adjusting for covariates. The associations of energy drink consumption with alcohol consumption, tobacco cigarette smoking and cannabis use were modified by grade level. To our knowledge, we are the first to show that the association between energy drink consumption and substance use is stronger among middle school students than their high school counterparts. Although males were significantly more likely to report energy drink consumption than females, the relationship between energy drink consumption and substance use outcomes did not differ by sex.

### 4.2. Comparison with Other Literature

Our results corroborate those from previous studies indicating that energy drink consumption among adolescents is associated with substance use [12,14]. In a sample of nearly 5000 10th grade Turkish students, Evren and Evren [37] found that energy drink consumption was associated with greater risk for lifetime tobacco, alcohol, and illicit drug use, with more frequent consumption having a stronger association with substance use. Similarly, Terry-McElrath et al. [16] found that the frequency of energy drink consumption is significantly and positively linked with the frequency of substance use in the past 30 days, such as alcohol, cigarettes, marijuana, and amphetamines, in a sample of over 20,000 USA adolescents in grades 8 to 12. These findings suggest that substance use behaviors tend to cluster within individuals [12,38]. Our study extend previous findings by showing that even though high school students were more likely than middle school students to report energy drink consumption, middle school students who consume energy drinks are at greater risk of also using alcohol, cannabis and tobacco cigarettes than high schoolers who consume energy drinks.

Beyond the adolescent-oriented marketing of energy drink companies [1,2,3], students consume energy drinks for several reasons. Energy drinks are often marketed as providing a quick boost of energy, which may attract the attention of adolescents who may be going through exams, sport season, and end of the school year party season. In a sample of adolescents aged 13–19 years utilizing emergency department services for any reason, Nordt et al. [39] identified several reasons for energy drink consumption, including to increase energy (61%), as a study aid (32%), to improve sports performance (29%), and to lose weight (9%). However, energy drink consumption can have negative effects on health, such as dehydration, arrythmia, heart failure, anxiety, and insomnia [40]. Research has further indicated that adolescents commonly consume energy drinks in combination with other drugs, particularly alcohol, often in an attempt to reduce the subjective feelings of alcohol intoxication [41]. The negative effects of such a combination can be devastating [42,43]. However, our survey did not measure energy drink consumption mixed with alcohol. Our results showed that 42.1% of adolescents who reported alcohol consumption in the past 12 months also reported energy drink consumption in the past 12 months, compared to only 18.6% of those who did not report alcohol consumption. There is also evidence of mixing energy drinks with cannabis and prescription drugs [39,40]. For example, Nordt et al. [39] found that 24% of adolescents who visited emergency room for any reason reported using energy drinks with ethanol or drugs, such as cannabis, cocaine, and methamphetamine.

In the current study, energy drink consumption was not associated with binge drinking among middle school students before and after adjusting for important covariates. It is possible that alcohol consumption among middle school students is not sufficiently frequent to result in binge drinking. As such, the association with binge drinking among middle school students might be more evident at higher levels of energy drink consumption for them (e.g., 5+ time in the past week). For example, Park et al. [44] found that adolescents who consumed energy drinks for 5 days or more on a weekly basis were at the greatest risk of mental health problems. Our measure of “at least one consumption in the past year” might be grouping more moderate and heavy users together. Future studies may consider exploring the frequency of consumption in relation to binge drinking among middle school students. Regardless, our results provide evidence of more propensity to substance use among middle school students who consume energy drinks. Future research is needed to replicate and further understand the grade level differences in the association between energy drink consumption and substance use among adolescents.

### 4.3. Strengths and Limitations

Strengths of this study include the large sample size and the use of different substance use indicators and sensitivity analyses using elevated levels of substance use as outcomes. We also used survey weights and Taylor series linearization methods to account for non-response bias and the complex study design, respectively. Nevertheless, this study is limited in several ways. First, given the cross-sectional nature of our data, causality assumptions should not be inferred about the observed relationship between energy drink consumption and substance use. Longitudinal studies are necessary to confirm temporality. Second, the data are self-reported and thus subject to response bias, particularly for more sensitive questions, such as those related to substance use. Third, the present study cannot determine whether alcohol consumption and energy drink consumption occurred at the same time, as the survey did not measure alcohol mixed with energy drink. Future studies are necessary to examine grade level differences in the consumption of alcohol mixed with energy drinks in relation with other drugs. Finally, the survey sampled students within Ontario’s publicly funded school systems and thus excluded approximately 8% of the Ontario student population. As such, the results are only generalizable to students in Grades 7 through 12 in publicly funded schools in Ontario. It may be that the excluded group of adolescents, mostly from private and alternative schools, differs with respect to energy drink consumption.

## 5. Conclusions

Our results clearly show that energy drink consumption is associated with substance use among adolescents, particularly among middle school students. Our results provide supporting evidence that middle school students who consume energy drinks are at higher risk of other substance use than their high school counterparts. Our results underscore the need to raise awareness and educate youth on the negative impacts of energy drink consumption. Teachers, parents, and health service providers should educate young students about the harmful adverse effects of consuming energy drinks. Future research using a longitudinal design is needed to replicate these findings and disentangle the observed grade level differences in the association between energy drink consumption and substance use among adolescents.

## Figures and Tables

**Table 1 ijerph-17-03110-t001:** Descriptive characteristics of the full sample and stratified by sex and grade level.

Characteristics	Total Sample *n* = 10,662	Males *n* = 4612	Females *n* = 6050	*p*-Value ^a^	Middle School *n* = 3498	High school *n* = 7164	*p*-Value ^a^
Total (%)	100	51.2	48.8		26.2	73.8	
Age (years)							
Mean (SD)	15.1 (1.8)	15.1 (1.7)	15.0 (1.9)	0.264	12.7 (0.8)	15.9 (1.2)	<0.001
Sex (%)							
Female	48.8				49.2	48.7	0. 859
Male	51.2				50.8	51.3	
Grade							
7	13.0	12.4	13.7	0.710	49.7	0	<0.001
8	13.2	13.7	12.7		50.3	0	
9	16.1	15.5	16.8		0	21.8	
10	17.0	17.5	16.5		0	23.0	
11	17.6	16.8	18.3		0	23.8	
12	23.1	24.2	22.0		0	31.4	
Ethnic background							
White	55.5	57.8	53.2	0.234	52.9	56.5	0.236
Black	10.3	9.7	11.0		9.1	10.8	
East/South-East Asian	8.8	8.4	9.3		8.2	9.0	
South Asian	6.9	7.0	6.8		9.0	6.2	
Other	18.4	17.1	19.8		20.8	17.6	
Subjective socioeconomic status							
Mean (SD)	6.9 (1.7)	6.9 (1.6)	6.9 (1.8)	0.934	7.2 (1.7)	6.8 (1.6)	<0.001
Screen time recommendation							
Not meeting	66.7	66.3	67.2	0.517	60.7	68.9	<0.001
Meeting (<2 h/day)	33.3	33.7	32.8		39.3	31.2	
Alcohol use							
No	84.8	83.7	86.1	0.995	59.7	25.0	<0.001
Yes	15.2	16.3	13.9		40.3	75.0	
Smoking tobacco cigarette							
No	78.5	78.0	79.1	0.143	97.4	80.4	<0.001
Yes	21.5	22.1	20.9		2.6	19.6	
Cannabis use							
No	83.1	82.3	83.9	0.364	97.3	71.9	<0.001
Yes	16.9	17.7	16.1		2.7	28.1	
Binge drinking							
No	89.3	89.6	88.9	0.165	98.4	77.6	<0.001
Yes	10.8	10.4	11.1		1.6	22.4	
Nonmedical use of prescribed opioids							
No	84.8	83.7	86.1	0.526	91.1	88.6	0.061
Yes	15.2	16.3	13.9		8.9	11.4	
Energy drink consumption in the past 12 months							
No	65.9	58.9	73.2	<0.001	76.3	62.2	<0.001
Yes	34.1	41.1	26.8		23.7	37.8	

Data are shown as weighted %, unless otherwise indicated. SD: standard deviation; SES: socioeconomic status. ^a^
*p*-Value of difference between males and females or middle and high school students based on a Pearson’s χ^2^-test transformed into an F-statistic for categorical data or an adjusted (for survey design) Wald test for continuous data.

**Table 2 ijerph-17-03110-t002:** Bivariate associations between energy drink consumption in the past 12 months, sociodemographic, and behavioral characteristics among adolescents.

Characteristics	% (95% CI)	*p*-Value ^a^
Total	34.1 (31.5–36.9)	
Age (years)		
Mean (SD)	15.4 (15.3–15.5)	<0.001
Sex (%)		
Female	26.8 (24.5–29.3)	<0.001
Male	41.1 (37.8–44.5)	
Grade		
7	21.2 (17.8–25.0)	<0.001
8	26.3 (22.0–31.1)	
9	36.9 (32.1–42.0)	
10	37.8 (30.8–45.3)	
11	36.6 (28.3–45.7)	
12	39.5 (35.0–44.1)	
Ethnic background		
White	35.6 (32.0–39.4)	0.002
Black	27.9 (23.6–32.7)	
East/South-East Asian	26.3 (22.5–30.5)	
South Asian	29.9 (25.3–34.9)	
Other	38.5 (33.3–43.8)	
Subjective socioeconomic status		
Mean (SD)	15.4 (15.3–15.5)	<0.001
Screen time		
Not meeting	35.4 (33.0–37.9)	0.010
Meeting	31.5 (27.7–35.6)	
Alcohol use in the last 12 months		
No	18.6 (16.8–20.6)	<0.001
Yes	42.1 (38.6–45.7)	
Smoking tobacco cigarette in the last 12 months		
No	27.7 (25.7–29.7)	<0.001
Yes	70.1 (64.9–74.9)	
Cannabis use in the last 12 months		
No	26.0 (23.7–28.4)	<0.001
Yes	63.9 (60.2–67.5)	
Binge drinking in the last 4 weeks		
No	28.7 (26.9–30.7)	<0.001
Yes	60.5 (53.2–67.4)	
Nonmedical use of prescribed opioid in the last 12 months		
No	31.9 (29.1–34.7)	<0.001
Yes	53.0 (47.5–58.4)	

Data are shown as weighted %, unless otherwise indicated. CI: confidence intervals; SD: standard deviation; SES: socioeconomic status. ^a^
*p*-Value of association with energy drink consumption based on a Pearson’s χ^2^-test transformed into an F-statistic.

**Table 3 ijerph-17-03110-t003:** Crude and adjusted Poisson regression models for the associations between energy drink consumption in the past 12 months and substance use outcomes among adolescents, OSDUHS, 2017 (*n* = 10,662).

Model	Alcohol Consumption	Smoking Tobacco Cigarette	Cannabis Use	Binge Drinking	Opioid Use
	IRR (95% CI)	IRR (95% CI)	IRR (95% CI)	IRR (95% CI)	IRR (95% CI)
**Model 1**	1.41 (1.34–1.47)	4.54 (3.81–5.40)	3.42 (3.06–3.82)	2.96 (2.40–3.65)	2.17 (1.78–2.66)
**Model 2**	1.31 (1.26–1.36)	3.74 (3.22–4.35)	2.90 (2.53–3.32)	2.46 (2.05–2.96)	2.23 (1.85–2.68)
**Model 3**					
Energy drink consumption × grade level	0.85 (0.77–0.93)	0.43 (0.24–0.80)	0.27 (0.15–0.46)	0.60 (0.13–2.65)	1.33 (0.86–2.07)
**Model 4**					
Energy drink consumption × sex	1.04 (0.98–1.10)	0.85 (0.58–1.25)	1.14 (0.93–1.41)	1.12 (0.85–1.48)	0.81 (0.56–1.17)

OSDUHS: Ontario Student Drug Use and Health Survey; IRR: incidence rate ratio; CI: confidence interval; NS: nonsignificant. Model 1 is unadjusted; Model 2 is adjusted for age, sex, ethnicity, subjective socioeconomic status, and screen time; Model 3 is Model 2 + interaction term between energy drink consumption and grade level; Model 4 is Model 2 + interaction term between energy drink consumption and sex.

**Table 4 ijerph-17-03110-t004:** Crude and adjusted Poisson regression models for the associations between energy drink consumption in the past 12 months and substance use outcomes stratified by grade level, OSDUHS, 2017 (*n* = 10,662).

Model	Alcohol Consumption	Smoking Tobacco Cigarette	Cannabis Use	Binge Drinking	Opioid Use
	IRR (95% CI)	IRR (95% CI)	IRR (95% CI)	IRR (95% CI)	IRR (95% CI)
**Middle school (*n* = 3498)**					
Model 1	1.53 (1.39–1.67)	8.58 (4.79–15.37)	10.72 (6.24–18.42)	4.09 (0.92–18.15)	1.79 (1.26–2.54)
Model 2	1.43 (1.30–1.57)	7.97 (4.43–14.33)	10.27 (5.63–18.75)	3.33 (0.65–17.11)	1.78 (1.23–2.58)
**High school (*n* = 7164)**					
Model 1	1.29 (1.24–1.35)	3.84 (3.25–4.54)	2.86 (2.55–3.19)	2.53 (2.11–3.04)	2.25 (1.74–2.92)
Model 2	1.28 (1.23–1.33)	3.54 (3.04–4.12)	2.73 (2.38–3.12)	2.39 (2.00–2.85)	2.35 (1.89–2.92)

OSDUHS: Ontario Student Drug Use and Health Survey; IRR: incidence rate ratio; CI: confidence interval. Model 1 is unadjusted. Model 2 is adjusted for age, sex, ethnicity, subjective socioeconomic status, and screen time.

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
