# Peer review of "Energy Drink Consumption and Substance Use among Middle and High School Students"

_ijerph, 2020, doi:10.3390/ijerph17093110_

Round 1
Reviewer 1 Report
The manuscript “Energy drink consumption and substance use among middle and high school students” is very well written, carefully analyzed, and cautiously interpreted.
General concerns
- Quality control measures are presumably in prior papers by this team. A brief summary of steps taken to identify participants that didn’t take the data collection seriously (responding A, B, C, D repeatedly or having past year but not lifetime use of a drug) and were excluded should be provided here too.
- In the interest of full transparency, if it does not contain any copyrighted items, consider including the full survey as a supplemental material.
- Mention of the IRB approval is needed.
Minor points
Line 4: Pubmed indexes authors by middle-initial
L 39: consider just a bit about the mechanism of action of caffeine and more specificity about which mechanisms (adenosine?) that it targets that impact brain development. If anything is known about sex differences in caffeine pharmacokinetics, that might form an important foundation for studying sex differences too. Perhaps a quick nod to the DSM and caffeine use disorder?
L 79: report percentages to the tenths (93.X%), l 167-8, 175 too but also throughout
L108: For future surveys, listing Demerol as an example opioid is odd as use, at least in the US, is extremely low.
l146: did you run a statistical comparison on excluded versus included? If so, a quick p > .05 would suffice.
L200: 12-months
L245: was
L323: Non-proper nouns in the article title should be in lower-case (L 406, 410, too).
Author Response
REVIEWER 1
Comments and Suggestions for Authors
The manuscript “Energy drink consumption and substance use among middle and high school students” is very well written, carefully analyzed, and cautiously interpreted.
General concerns
- Quality control measures are presumably in prior papers by this team. A brief summary of steps taken to identify participants that didn’t take the data collection seriously (responding A, B, C, D repeatedly or having past year but not lifetime use of a drug) and were excluded should be provided here too.
RESPONSE: To maximize validity and to enhance cross-study comparability, many of the OSDUHS questionnaire items were derived from international guidelines and recognized student surveys such as NIDA’s Monitoring the Future (MTF) survey, the CDC’s Youth Risk Behavior Survey (YRBS), and the WHO’s Health Behaviour in School-aged Children (HBSC) survey, and have been shown to produce valid responses. Furthermore, the survey uses validated scales and screeners. All newly introduced items in the 2017 questionnaire were evaluated by both expert review and pretested on a small convenience sample of young adolescents. The readability of the 2017 questionnaire showed a 7th-grade reading level according to the Flesch-Kincaid reading score. At the end of the questionnaire students were asked to evaluate the comprehension and sensitive nature of the questionnaire. The majority of students indicated positive assessments: 97% of students (96% of 7th graders) indicated that the questionnaire was “fairly” or “very easy” to understand; only 10% of students (7% of 7th graders) indicated that the questionnaire was “much too long”; and only 5% of students (6% of 7th graders) indicated that questions in the survey would make most students “very uncomfortable.” This latter finding provides some reassurance that social desirability should not greatly bias our estimates, even among the youngest students. We have added a brief description to the Methods section.
- In the interest of full transparency, if it does not contain any copyrighted items, consider including the full survey as a supplemental material.
RESPONSE: We have added a link to the survey to our paper.
- Mention of the IRB approval is needed.
RESPONSE: We have done so.
Minor points
Line 4: Pubmed indexes authors by middle-initial
RESPONSE: Authors list has been updated.
L 39: consider just a bit about the mechanism of action of caffeine and more specificity about which mechanisms (adenosine?) that it targets that impact brain development. If anything is known about sex differences in caffeine pharmacokinetics, that might form an important foundation for studying sex differences too. Perhaps a quick nod to the DSM and caffeine use disorder?
RESPONSE: We have added a bit more on the mechanism of action of caffeine and observed sex differences to the revised version of the manuscript.
L 79: report percentages to the tenths (93.X%), l 167-8, 175 too but also throughout
RESPONSE: We have done so.
L108: For future surveys, listing Demerol as an example opioid is odd as use, at least in the US, is extremely low.
RESPONSE: We thank the reviewer for this comment. We will keep it in mind for future surveys.
l146: did you run a statistical comparison on excluded versus included? If so, a quick p > .05 would suffice.
RESPONSE: Yes, we did so, and reported the results in the statistical analysis section.
L200: 12-months
RESPONSE: We corrected that.
L245: was
RESPONSE: We have changed “is” to “was”.
L323: Non-proper nouns in the article title should be in lower-case (L 406, 410, too).
RESPONSE: Changes were made as suggested by the reviewer.

Reviewer 2 Report
Referee Report on: “Energy Drink Consumption and Substance Use among Middle and High School Students”
There are three main concerns about the current version of the paper: (1) there is not a true contribution to the literature, (2) school type is incorrectly defined, and (3) the discussion of energy drinking in the last 7 days is confusing (at best).
The authors argue that the main contribution of the paper is that previous studies have focused more on grades and sex, but not on school type. However, the way that the authors define school type relies on aggregated different school grades into middle school and high school. In essence, the authors examine a similar relationship as previous studies, but instead of examining separate school grades, they aggregate them into school levels. Nevertheless, these are not school types. A school type refers to classification into public, private, charter, special needs, international, and so on. On page 2, the authors exclude all such school types from their analysis apart from public schools. The authors would need to reconsider what their contribution to the literature is and adjust the writing accordingly. Claiming that they examine the association for different school types is misleading and erroneous. If their goal is indeed on school types, then the whole statistical analysis needs to be reconsidered since it includes only one school type.
The authors should be much clearer about the energy consumption variable. Page 3 emphasizes on this measure as consumption during the last 7 days, and only at the end of the paragraph in 2.2.2., the authors add “last 12 months.” Is this measure 1 if they had some consumption in the last 12 months, and 0 otherwise? Why is there a discussion about consumption in the last 7 days? If all the outcome variables are measured as consumption in the last 12 months, we cannot rely on examining how energy drink consumption in the last 7 days is associated with consumption of other substances during the last 12 months. By construction of the questionnaire, consuming energy drinks in the last 7 days cannot affect consumption of other substances in the last 12 months. Despite this timing inconsistency, the authors still discuss energy drink consumption in Tables 1 and 2, but then not run regression results for this measure. I would recommend removing any discussion about energy drink consumption during the last 7 days from the paper. The authors could use it as a falsification test after they present their main findings about the association of energy drink consumption with other substance consumption with both of them measured at the same period of 12 months. Should they estimate the association with energy consumption during the last 7 days, they should find insignificant associations with the other substance uses.
Some secondary comments:
- Aren’t age and grades highly colinear when included in the mode?
- Add a sentence or two explaining the need to use a Poisson model.
- There is hardly any discussion about the suggested “moderators.”
Author Response
REVIEWER 2
Comments and Suggestions for Authors
Referee Report on: “Energy Drink Consumption and Substance Use among Middle and High School Students”
There are three main concerns about the current version of the paper: (1) there is not a true contribution to the literature, (2) school type is incorrectly defined, and (3) the discussion of energy drinking in the last 7 days is confusing (at best).
The authors argue that the main contribution of the paper is that previous studies have focused more on grades and sex, but not on school type. However, the way that the authors define school type relies on aggregated different school grades into middle school and high school. In essence, the authors examine a similar relationship as previous studies, but instead of examining separate school grades, they aggregate them into school levels. Nevertheless, these are not school types. A school type refers to classification into public, private, charter, special needs, international, and so on. On page 2, the authors exclude all such school types from their analysis apart from public schools. The authors would need to reconsider what their contribution to the literature is and adjust the writing accordingly. Claiming that they examine the association for different school types is misleading and erroneous. If their goal is indeed on school types, then the whole statistical analysis needs to be reconsidered since it includes only one school type.
RESPONSE: We thank the reviewer for this comment. To avoid confusion, we have changed “school types” to “levels of education”. We collapsed the grades into two groups for two reasons. First, school intervention programs related to energy drink consumption focus on high school students, as they have higher prevalence of consumption than their middle school counterparts. However, we believe that consumption in younger students is more devastating because young children still have their brain under development. Thus, contrasting the two groups provides valuable information for future research by generating hypothesis and for future interventions intended to decrease consumption of energy drinks and substance use among schoolchildren. Second, collapsing school grades into two groups rather than treated them individually help increase statistical power. We have revised the introduction section accordingly and we still believe that the present paper is original and adds an important contribution to the knowledge base.
The authors should be much clearer about the energy consumption variable. Page 3 emphasizes on this measure as consumption during the last 7 days, and only at the end of the paragraph in 2.2.2., the authors add “last 12 months.” Is this measure 1 if they had some consumption in the last 12 months, and 0 otherwise? Why is there a discussion about consumption in the last 7 days? If all the outcome variables are measured as consumption in the last 12 months, we cannot rely on examining how energy drink consumption in the last 7 days is associated with consumption of other substances during the last 12 months. By construction of the questionnaire, consuming energy drinks in the last 7 days cannot affect consumption of other substances in the last 12 months. Despite this timing inconsistency, the authors still discuss energy drink consumption in Tables 1 and 2, but then not run regression results for this measure. I would recommend removing any discussion about energy drink consumption during the last 7 days from the paper. The authors could use it as a falsification test after they present their main findings about the association of energy drink consumption with other substance consumption with both of them measured at the same period of 12 months. Should they estimate the association with energy consumption during the last 7 days, they should find insignificant associations with the other substance uses.
RESPONSE: The item was measured using the following question: “In the last 7 days, how often did you drink a can of a high-energy caffeine drink, such as Red Bull, Rockstar, Amp, Full Throttle, Monster, etc.?” Response categories referred to frequent consumption (differentiating between “1 time”, “2 to 4 times”, “5 to 6 times”, “once each day”, or “more than once each day”); no consumption in the last 7 days, but some consumption in the last 12 months; and no consumption in the last 7 days or in the last 12 months. Following previous publications, we constructed a dichotomous measure to represent “no consumption in the last 7 days, but some consumption in the last 12 months” versus “non-consumption of energy drinks in the last 12 months”. We have added some references to the manuscript. We have also deleted the use of consumption in the last 7 days as an outcome throughout the paper, as recommended by the reviewer (i.e. Methods and Results sections, and Tables 1 and 2).
Some secondary comments:
- Aren’t age and grades highly colinear when included in the mode?
RESPONSE: There was no collinearity between age and grades in this sample. We have added this to the statistical analysis section.
- Add a sentence or two explaining the need to use a Poisson model.
RESPONSE: We used Poisson regression following the recommendation to estimate relative risk for a common (i.e., prevalence of 10 % or more) binary outcome (Cummings, 2009, Zou, 2004). Poisson regression is more conservative than logistic regression and it was more desirable to estimate a relative risk or risk ratio (RR) instead of an odds ratio (OR) since there is an increasing differential between the RR and OR with increasing prevalence rates. We have added this justification to the text as well as relevant references (Cummings, 2009, Zou, 2004).
- There is hardly any discussion about the suggested “moderators.”
RESPONSE: We have added a discussion about the suggested moderators.
REFERENCES
Cummings, P., 2009. Methods for Estimating Adjusted Risk Ratios. The Stata Journal, 9(2), 175–196.
Zou, G., 2004. A modified poisson regression approach to prospective studies with binary data. Am J Epidemiol. 159(7), 702-706.

Round 2
Reviewer 2 Report
Referee Report on: “Energy Drink Consumption and Substance Use among Middle and High School Students”
There are three main concerns about the current version of the paper: (1) there is not a true contribution to the literature, (2) school type is incorrectly defined, and (3) the discussion of energy drinking in the last 7 days is confusing (at best).
The authors argue that the main contribution of the paper is that previous studies have focused more on grades and sex, but not on school type. However, the way that the authors define school type relies on aggregated different school grades into middle school and high school. In essence, the authors examine a similar relationship as previous studies, but instead of examining separate school grades, they aggregate them into school levels. Nevertheless, these are not school types. A school type refers to classification into public, private, charter, special needs, international, and so on. On page 2, the authors exclude all such school types from their analysis apart from public schools. The authors would need to reconsider what their contribution to the literature is and adjust the writing accordingly. Claiming that they examine the association for different school types is misleading and erroneous. If their goal is indeed on school types, then the whole statistical analysis needs to be reconsidered since it includes only one school type.
AUTHORS’ RESPONSE: We thank the reviewer for this comment. To avoid confusion, we have changed “school types” to “levels of education”. We collapsed the grades into two groups for two reasons. First, school intervention programs related to energy drink consumption focus on high school students, as they have higher prevalence of consumption than their middle school counterparts. However, we believe that consumption in younger students is more devastating because young children still have their brain under development. Thus, contrasting the two groups provides valuable information for future research by generating hypothesis and for future interventions intended to decrease consumption of energy drinks and substance use among schoolchildren. Second, collapsing school grades into two groups rather than treated them individually help increase statistical power. We have revised the introduction section accordingly and we still believe that the present paper is original and adds an important contribution to the knowledge base.
REFEREE COMMENTS: The authors adjusted the terminology from school types to level of education, however, this is still not accurate. Reference to levels of education suggests that the paper is about high school vs. some college vs. college vs. more than college. This is not the goal of the paper since the authors focus on mandatory education and they examine grade levels rather than levels of education in general.
More importantly, though, the comment on the contribution of the paper has not been addressed. If the paper is indeed about grade levels, how is it different from earlier studies that have already examined the question for middle school and high school students? The last paragraph in the Introduction argues that there is lack of research on whether/how gender and grade level may be related to energy drink consumption and substance use. This is in contrast with section 4.2 where previous studies are compared to the current study for the same topic and the same grade level. I would encourage the authors to carefully think of the value added of their paper. It seems that their marginal contribution is about the use of middle school students in relation to high school students. But it should be clearer why such a distinction is important and how it is different from previous studies that have examined students in grade 8-12.
The authors should be much clearer about the energy consumption variable. Page 3 emphasizes on this measure as consumption during the last 7 days, and only at the end of the paragraph in 2.2.2., the authors add “last 12 months.” Is this measure 1 if they had some consumption in the last 12 months, and 0 otherwise? Why is there a discussion about consumption in the last 7 days? If all the outcome variables are measured as consumption in the last 12 months, we cannot rely on examining how energy drink consumption in the last 7 days is associated with consumption of other substances during the last 12 months. By construction of the questionnaire, consuming energy drinks in the last 7 days cannot affect consumption of other substances in the last 12 months. Despite this timing inconsistency, the authors still discuss energy drink consumption in Tables 1 and 2, but then not run regression results for this measure. I would recommend removing any discussion about energy drink consumption during the last 7 days from the paper. The authors could use it as a falsification test after they present their main findings about the association of energy drink consumption with other substance consumption with both of them measured at the same period of 12 months. Should they estimate the association with energy consumption during the last 7 days, they should find insignificant associations with the other substance uses.
AUTHORS’ RESPONSE: The item was measured using the following question: “In the last 7 days, how often did you drink a can of a high-energy caffeine drink, such as Red Bull, Rockstar, Amp, Full Throttle, Monster, etc.?” Response categories referred to frequent consumption (differentiating between “1 time”, “2 to 4 times”, “5 to 6 times”, “once each day”, or “more than once each day”); no consumption in the last 7 days, but some consumption in the last 12 months; and no consumption in the last 7 days or in the last 12 months. Following previous publications, we constructed a dichotomous measure to represent “no consumption in the last 7 days, but some consumption in the last 12 months” versus “non-consumption of energy drinks in the last 12 months”. We have added some references to the manuscript. We have also deleted the use of consumption in the last 7 days as an outcome throughout the paper, as recommended by the reviewer (i.e. Methods and Results sections, and Tables 1 and 2).
REFEREE COMMENTS: Similar, to the comment above, the suggestion was only partially addressed. Did the authors conduct the suggested falsification test for energy drink consumption during the last 7 days, and what was the association with the other substance uses? Given that this is a correlation study, the authors should present more robustness analysis of their results.
Some secondary comments:
- There is hardly any discussion about the suggested “moderators.”
- AUTHORS’ RESPONSE: We have added a discussion about the suggested moderators.
- REFEREE COMMENTS: Unfortunately, I still do not see a sufficient discussion about why these are treated as moderators. The current structure only indicates that behaviors of middle and high school students should be analyzed separately but this does not qualify them as moderators.
Author Response
Comments and Suggestions for Authors
Referee Report on: “Energy Drink Consumption and Substance Use among Middle and High School Students”
There are three main concerns about the current version of the paper: (1) there is not a true contribution to the literature, (2) school type is incorrectly defined, and (3) the discussion of energy drinking in the last 7 days is confusing (at best).
REFEREE COMMENTS: The authors adjusted the terminology from school types to level of education, however, this is still not accurate. Reference to levels of education suggests that the paper is about high school vs. some college vs. college vs. more than college. This is not the goal of the paper since the authors focus on mandatory education and they examine grade levels rather than levels of education in general.
AUTHORS’ RESPONSE: Following the suggestion, we have changed “level of education” to grade level.
More importantly, though, the comment on the contribution of the paper has not been addressed. If the paper is indeed about grade levels, how is it different from earlier studies that have already examined the question for middle school and high school students? The last paragraph in the Introduction argues that there is lack of research on whether/how gender and grade level may be related to energy drink consumption and substance use. This is in contrast with section 4.2 where previous studies are compared to the current study for the same topic and the same grade level. I would encourage the authors to carefully think of the value added of their paper. It seems that their marginal contribution is about the use of middle school students in relation to high school students. But it should be clearer why such a distinction is important and how it is different from previous studies that have examined students in grade 8-12.
AUTHORS’ RESPONSE: We thank the reviewer for this comment. We have revised the introduction and indicated that previous research has either focused on high school students or combined both middle and high school students in their analyses. Using middle school students and compare them to high school students is important for several reasons. First, it helps to demonstrate whether the associations between energy drink consumption and substance use vary between middle and high school students (and they do). The day-to-day life of these two groups is quite different and it is reasonable to assume that the associations would differ. Second, it is important to know whether there is a need for intervention programs to start early on (i.e. target middle school students rather than their high school counterparts only). Finally, it helps to inform future research regarding the need or not to be conducted using middle school students in relation to high school students.
REFEREE COMMENTS: Similar, to the comment above, the suggestion was only partially addressed. Did the authors conduct the suggested falsification test for energy drink consumption during the last 7 days, and what was the association with the other substance uses? Given that this is a correlation study, the authors should present more robustness analysis of their results.
AUTHORS’ RESPONSE: We thank the reviewer for this comment. We have tested the association between energy drink consumption during the last 7 days and substance use outcomes and found that it is significantly associated with each of the outcomes in both middle and high school students. We have chosen to not include it to avoid confusion given the timing inconsistency between exposure and the outcomes.
REFEREE COMMENTS: Unfortunately, I still do not see a sufficient discussion about why these are treated as moderators. The current structure only indicates that behaviors of middle and high school students should be analyzed separately but this does not qualify them as moderators.
AUTHORS’ RESPONSE: We have revised the introduction and added a justification on why sex and grade level should be treated as moderators.